# Exercise to Improve Postural Stability in Older Adults with Alzheimer’s Disease: A Systematic Review of Randomized Control Trials

**DOI:** 10.3390/ijerph191610350

**Published:** 2022-08-19

**Authors:** Mohamad Asyraf Adzhar, Donald Manlapaz, Devinder Kaur Ajit Singh, Normala Mesbah

**Affiliations:** 1Physiotherapy Program & Centre for Healthy Ageing & Wellness, Faculty of Health Sciences, Universiti Kebangsaan Malaysia, Kuala Lumpur 50300, Malaysia; 2Department of Physical Therapy, College of Rehabilitation Sciences, University of Santo Tomas, Manila 1008, Philippines

**Keywords:** exercise, physical activity, falls, postural instability, Alzheimer’s disease

## Abstract

In this systematic review, we aim to synthesize published evidence on the effects of exercise in improving postural stability among older adults with Alzheimer’s Disease (AD). A systematic electronic literature search was undertaken in Pedro, Cochrane, MEDLINE, ProQuest, Science direct and Clinical trial databases involving human participants published from year 2000–2022. This search was updated in June 2022. The studies chosen were based on predetermined criteria. Data relating to the contents and parameters of exercise in persons with AD were gathered and analyzed. A total of 8 experimental studies met the inclusion criteria. Overall, the selected studies were of a medium quality. In these studies, information and physical exercises were used to improve postural stability in older adults with AD. The findings of the review suggest that performing combined strength, balance and executive function training can improve postural stability. However, we are unable to conclude the specific dose for specific type of exercise. More high-quality studies are required pertaining to exercise prescription for older adults with AD. Mostly, information and physical exercise were delivered via face-to-face sessions conducted by health professionals. The structure of exercises summarized in this review may be beneficial for older adults with AD to improve postural stability and as a result reduce falls.

## 1. Introduction

Alzheimer’s disease (AD) is a deadly neurodegenerative disease that causes progressive deterioration of memory and self-awareness in cognitive, neuropsychiatric, and functional manner [1]. In year 2005, 24.2 million people worldwide had dementia with 4.6 million new cases being diagnosed yearly, accounting for over 70% of AD cases. AD prevalence and incidence rates rise exponentially with age [2]. Falls occur at a rate of 0.65 per person per year in community-dwelling older adults and rise to 1.7 per person per year for older adults in residential aged care (RAC) [3].

In year 2015, $818 billion (USD) were spent on treating AD according to AD International. This represents an increase of 35% from year 2010. This numberis projected to reach $2 trillion (USD) by year 2030 [4]. The total cost of treatment for older adults with AD increased with AD severity [5]. In addition, the increasing dependency of older adults with AD on their caregivers also raises the expenses of informal care in the later stages of the disease [6].

It has been reported that older people with cognitive impairment (CI) are twice as likely to fall and be injured compared to those without CI [7,8]. Factors such as frailty, gait impairments and CI contribute to falls, fractures, injuries, and even mortality [5,7]. Gait and cognitive impairments are linked, indicating that gait is no longer just a motor task but rather an activity that involves executive function, attention, and the environment [7,9,10]. Approximately 50% of older adults with AD have difficulty walking three years following diagnosis, with 33% of these adults classified as non-ambulatory [11].

Hip fractures because of falls may have a negative impact on functional capacity and survival in older people, particularly those with CI. They tend to have a poorer prognosis than those without CI [12]. In older adults with AD, exercise has been shown to improve their functional performance and physical health [13]. Exercise programs that challenge postural balance and are performed at least three times per week have been shown to be most effective in preventing falls [14]. Furthermore, evidence supports 3 months of multicomponent exercise program (strength, endurance, balance, aerobic, flexibility and dual tasking) on falls prevention in older adults with CI [9,12,13,15,16]. As AD progresses, physical activity may help prevent or delay the development of severe neuropsychiatric symptoms [17]. In the Finnish Alzheimer Disease Exercise (FINALEX) study, intensive exercise intervention delayed regression in physical function in older adults with AD in the long term and decreased the risk of falling [18]. It has also been shown that physical activity is cost-effective in slowing the progression of AD in the long-term [5].

Furthermore, exercise improves cognitive status in older adults with AD and has other positive effects on physical function, such as reducing falls, increasing autonomy and quality of life [5]. Therefore, in this systematic review, we aimed to synthesize published evidence on the effects of exercise in improving postural stability in older adults with AD.

## 2. Materials and Methods

This review is prospectively registered with the International Prospective Register of Systematic Reviews (PROSPERO: CRD42021275131). The methodology in this review is reported based on the Cochrane for Systematic Reviews of Intervention. The Preferred Reporting Items for Systematic Reviews and Meta-Analyses (PRISMA) were followed.

### 2.1. Search Strategy

A detailed electronic search was conducted to identify English-language articles involving human participants published from year 2000 to 2022. The search was updated in June 2022. An advanced search index using combinations of three key terms generated search combinations. The titles and abstracts of these references were examined. Five databases, namely Pedro, Cochrane, MEDLINE, Science direct and Clinical trial database, were systematically searched. The scope of the search was constructed using the inclusion and exclusion criteria. Similar keywords were used to explore the databases.

### 2.2. Eligibility Criteria

Experimental studies involving participants aged 60 years and above who had been clinically diagnosed with AD were included. Cognitive function in the studies was measured using established tools such as AD Assessment Scale-Cognitive (ADAS-Cog11) or Mini-Mental State Examination (MMSE). All studies, regardless of the results, were included in the review. Studies that included balance exercises (any developed intervention requiring balance training) and outcome measures to measure balance (static and dynamic balance) were included. The interventions using balance exercises without a control group were excluded.

### 2.3. Selection of Studies

A preliminary screening of titles and abstracts was first undertaken. Full-text articles of potentially relevant studies were then retrieved and assessed in accordance with the established criteria. Finally, relevant studies were also identified by reviewing the references of the selected papers.

### 2.4. Methodological Quality Assessment

The methodological quality of the included studies was assessed using the PEDro scale assessment [19]. As the PEDro scale had been developed primarily to evaluate quantitative studies, it is an ideal choice for this evaluation. The ten significant components examined in the PEDro scale are random allocation, concealed allocation, groups similar at baseline, participant blinding, therapist blinding, assessor blinding, <15% dropouts, intention-to-treat analysis, reporting of between-group difference, the point estimate and reporting of variability. One point is given to each component with a clear explanation. The cumulative points were then used to measure the quality of reporting, with a higher score indicating higher methodological quality. The maximum score was ten. Two researchers (DKAS) and (DM) assessed the methodological quality of the studies, with discrepancies resolved through discussion.

### 2.5. Data Extraction and Synthesis

The characteristics of the exercises were extracted and summarized to provide the basis for comparison. A customized data extraction table of included studies was used to summarize the information about the contents and parameters used in the exercises. One researcher (MAA) extracted the data, and another researcher (NM) checked the data. The findings were synthesized using a narrative synthesis.

## 3. Results

### 3.1. Search Results

295 studies were identified from five database engines. The full-text articles of 73 studies were retrieved and assessed for eligibility. 65 studies were excluded as the average age of the sample population was less than 60 years (n = 2), and the intervention provided was not balance exercises (n = 63). A total of eight papers met our selection criteria and were chosen for this evaluation. The studies included in this review are as in the PRISMA flow diagram (Figure 1).

### 3.2. Methodological Quality of Included Study

The methodological quality of the included studies is summarized in Table 1 All included studies were randomized control trials [7,12,18,20,21,22,23,24]. The cumulative PEDro scores varied from low (score of 3) to medium quality (score of 7). Randomization was carried out in 6 studies [7,12,20,21,22,24], concealed allocation of randomization occurred in 3 studies [12,20,21], assessor blinding in 5 studies [12,20,22,23,24], intention-to-treat analysis in 5 studies [12,18,20,21,24], therapist blinding in 2 studies [12,24] and ≤15% dropouts in 3 studies [21,23,24]. There were no studies in which participants were blinded because of the difficulties associated with the interventions.

### 3.3. Study Characteristics

Table 2 depicts a summary of the included studies. The mean age of participants was 69.3 years, ranging from 60 to 95 years old [7,12,18,20,21,22,23,24]. Participants included in the study were diagnosed with mild and moderate CI using AD Assessment Scale-Cognitive 11 (ADAS-Cog 11) [7,20] with a mean score of 18.9 (SD = 8.5) and using Mini-Mental State Examination (MMSE) [12,18,21,22,23,24] with a mean score of 15.8 (SD = 5.3).

### 3.4. Outcome Measures to Evaluate Cognitive Function

Montreal Cognitive Assessment (MoCA) was used in one study with a baseline group score of 18 (SD = 5.6) and an intervention score of 16 (SD = 4.2) (Table 2). Clinical Dementia Rating (CDR) was used in two studies [18,21] with a mild dementia baseline group score of 22 (SD = 31.4) and intervention score of 23.5 (SD = 33.6). In one study, means ± standard deviations were not stated [18]. Mini-Mental State Examination (MMSE) was used in six studies [18,20,21,22,23,24], with a baseline group score of 11.6 (SD = 3.8) and an intervention score of 11.5 (SD = 3.7). Standardized Mini-Mental State Examination (sMMSE), with a baseline group score of 21.6 (SD = 4.6) and intervention score of 22.0 (SD = 4.7), was used in one study [20]. Quality of Life-AD (QOL-AD) was used in two studies [7,20], with a baseline group score of 38.5 (SD = 5.5) and an intervention score of 38.0 (SD = 6.0). AD Assessment Scale-Cognitive (ADAS-Cog) was administered in two studies [7,20] with a baseline group score of 29.3 (SD = 9.1) and an intervention score of 19.4 (SD = 7.8). 15-items Geriatric Depression Scale was used in two studies [7,12], with a baseline group score of 3.7 (SD = 3.2) and an intervention score of 3.4 (SD = 3.1). Montgomery-Asberg Depression Rating Scale (MADRS) was used in one study [24]^,^ with a baseline group score of 12.3 (SD = 6.0) and an intervention score of 11.9 (SD = 6.1).

### 3.5. Outcome Measures to Evaluate Postural Instability

Table 3 shows the outcome measures when evaluating postural stability. The outcome measure used to measure balance included Brief Balance Evaluation Systems Test (Brief BESTest) (one study) [7], Short Form Physical Performance Battery (SPPB) (two studies) [7,21] and Berg Balance Scale (one study) [12]. Baseline mean group score was 8.5 (SD = 5.5) for Brief BESTest and 7.6 (SD = 2.1) for SPBB. Meanwhile, the intervention mean group score was 9.0 (SD = 6.0) for Brief BESTest and a mean score of 7.9 (SD = 3.1) for SPBB. The 6-min walk test (6MWT) (two study) [20,22] and Functional Independence Measure (FIM) (two studies) [18,21] were also used. The baseline distance group mean score was 345.3 (m) (SD = 171.9) for 6MWT, and the mean score for FIM was 94.4 (SD = 13.0). Meanwhile, the intervention group’s mean score was 329.22 (m) (SD = 172.4) for 6MWT, and the mean score of FIM was 96.0 (SD = 14.0). The 6-min walking speed (one study) [24] baseline speed group mean score of 0.33 (ms^−1^) (SD = 0.14) and intervention mean score was the same, 0.33 (ms^−1^) (SD = 014). Timed up and go (TUG) (one study) [24] test baseline mean score was 2.70 (SD = 0.8) and intervention mean scored 3.05 (SD = 1.1) (time completed was scored as 1 = no instability to 5 = very abnormal). While, Tinetti score (one study) [23] baseline mean score was 17.0 (SD = 3.0) and intervention mean score was 22.0 (SD = 3.0).

### 3.6. Intervention for Older Adults with AD

Table 4 is a summary of the interventions included in the reviewed studies. Balance exercise was found in seven studies [7,12,18,21,22,23,24]. Aerobic exercise was also found in seven studies [7,18,20,21,22,23,24] and strength exercise in eight studies [7,12,18,20,21,22,23,24]. Flexibility exercise were used in three studies [22,23,24]. Other than that, executive function training [21] and dual-tasking exercises [20] were used in one study, respectively. The time for each exercise session varied from 15 to 90 min with 8 to 20 repetitions. The intensity varied from 2 to 5 times exercise per week. Furthermore, the total exercise duration was between 7 weeks to 12 months. In seven studies group exercises were performed [7,12,18,21,22,23,24], and in one study, individually tailored exercise was carried out [21].

In the control group, participants in five studies received interventions consisting of Cognitive Stimulation Therapy (CST) [7], written and verbal advice on exercise and nutrition intake [21], and structured activities (discussed around themes such as holidays and seasons) [12], counseling and advice on physical activity [20], and one-on-one conversations on therapeutically oriented interaction [22]. The control group in three studies did not receive any intervention besides routine nursing and medical care [18,23,24].

## 4. Discussion

Alzheimer’s disease (AD) is a neurodegenerative disorder in which memory, thinking capability and the ability to carry out the activities of daily living (ADL) deteriorate [25]. Most ADL may still be done partly, poorly or not at all. AD is known to affect the balance of the people who suffer from it. Compared to older adults in general, those with cognitive impairment (CI) have a higher risk of falling and sustaining falling-related injuries. The associated physical and cognitive impairments may be responsible for the increased risk [12].

Gait changes may be associated with AD and may precede cognitive changes [5]. Mental processing such as executive function, focus, and visuospatial perception are all influenced by cognitive impairments in older adults with AD [26]. Changes in even one either executive function, focus, or visuospatial perception will result in falls. Gait changes occur in 50% of persons with AD three years after diagnosis, and 33% of these persons lose their ability to walk [5]. In addition, a person with a Mini-Mental State Examination (MMSE) score of between 13 to 24 points is classified as mild (13 to 20 points) or moderate (21 to 24 points) CI [5]. A lower MMSE score is associated with a higher risk of falling [18].

Caution must be considered when conducting an exercise intervention with people with AD as there is a serious risk of falls. The results of the cognitive stimulation therapy (CST) with fall prevention exercise (CogEx) feasibility study depicted that even though the fall prevention exercise can be integrated into the CST, the combined program’s fidelity was low and other aspects of the study’s findings were inconclusive [7]. The results of this review suggest that future studies should consider providing trained facilitators during the CogEx training that is packaged with more guidance. It may also be physiotherapists led to deliver the falls prevention segment in the CogEx training package.

The effects of exercise that includes improving muscle mass and cardiovascular health, lowering inflammation and oxidative stress, and increasing brain plasticity, may lead to improved quality of life of persons with AD [27]. The American College of Sports Medicine (ACSM) recommends 150 min of moderate or 75 min of intense physical activity each week, split into 3 to 5 sessions. Persons with AD should engage in strengthening exercises two or more days per week, with a 3/2 aerobic/strength exercise ratio per week [5]. But another study showed that aerobic exercise twice a week for three months improved physical performance in persons with AD [18]. In addition, exercise on its own demonstrated a positive impact on physical function (e.g., fitness, muscle strength and balance) but not fall prevention [12]. However, combined cognitive training and exercise has been shown to assist person with AD to avoid falling [5,12].

Performing integrated strengthening, balance and mobility exercises improved stability in older persons [12]. For example, older persons were able to extend both upper and lower limbs or take an additional step to reach for an object with this multicomponent exercise intervention [12]. While exercises have been shown to improve physical function that may improve stability, the SPBB was challenging to test stability of older persons with AD due to difficulties in giving instructions to persons with dementia [21].

Finally, structured physical activity can help older adults in various stages of AD. While physical functioning may benefit older adults with moderate CI, fall prevention may benefit those with advanced dementia. Thus, the opportunity to minimize the impairment burden in older adults with AD should not be passed up. Whether performed in combination or separately, strength, resistance, and balance training have been shown to benefit older adults with various symptoms related to AD.

Pertaining to exercise dose (FITT: frequency, intensity, type and time), it is inconclusive. In the community settings [18,20,21], only one study [20] reported the FITT. Participants in this study [20] showed improvement in 6 min walking distance after 1 h, 2x/week for 12 months moderate to high intensity aerobic and strengthening exercise. However, high intensity exercises were not recommended as cognitive function may decline over time due to high intensity induced inflammation [28] and slow oxygenation in cortical area [29]. Whereas, four studies that were conducted in long term care provided FITT information [12,22,23,24]. The frequency varied from 2 to 5 times per week, time from 15 to 75 min per day, intensity from low to high for the duration of 3 to 12 months with inclusion of flexibility, strength, balance and aerobic exercises. Among all of the studies that showed improvement in balance, study by Roach et al. (2011) had the least amount of time per week 75−150 min, low to high intensity, and a four months intervention. Still, this dosage should be applied with precaution as the effects of exercise on cognitive function were not reported. We could not evaluate the specific FIT for specific type of exercise due to the fact that there were only a small number of related studies, difference in the FITT used, and lack of specific evaluation of its associated outcomes in the included studies. However, regardless of the cognitive functional status among more frail older adults with AD, the desired effect of the exercise is to improve or maintain mobility and ability to perform basic activities of daily living as long as possible. As such, individually tailored exercise prescription by a physiotherapist or recommended dose by WHO for older adults with sedentary behavior could be applied [30].

In addition, by adhering to the time and intensity prescription, it may be substantial for older adults to improve their muscle strength, balance and agility [31]. Aside from that, it enables the body to push the limits just enough to make progress without injuries. Thus, the effectiveness of the exercise can be achieved. Regarding the frequency of the exercises, 8 to 10 repetitions are recommended, but it can be varied, tailored to older adults’ conditions. Increasing frequency can be a progression in the exercise program. As a result, exercise can be regarded a beneficial intervention for improving functional capacity that includes muscle strength, gait, balance, mobility and as well as executive function, thus lowering the risk of falling [18].

### Limitation of the Review

One of the limitations of this study is limited current scientific evidence on exercise to improve postural stability and consequently reduce falls in older adults with AD. Moreover, most randomized control trials (RCTs) of exercise program for older persons institutionalized with CI were of short duration [13]. In addition, in all eight studies the outcome measures varied as for usual studies of complex treatments. Next, the validity of the results may have been reduced as the data from one study could not be retrieved. It has also been proposed that control groups may provide ‘hidden’ information about the likelihood of outcomes in randomized trials. Finally, potential data may have been left out of the current review.

## 5. Conclusions

Although in five randomized studies, the exercise on postural stability and reduction falls in older adults with AD were examined, only four of the studies are of good quality. The good quality studies nonetheless showed that a functional exercise program as a single intervention does not prevent falls in persons with CI [12], and physical exercise improves physical fitness in the short term. In addition, multi-domain interventions (physical exercise with executive function and dual-task training) with intense and sufficient duration in older adults with AD improve their mobility and physical function [18,21]. Similarly, cognitive stimulation therapy with fall prevention exercise improves balance of older adults with CI [7]. This review provides insight to the practitioner and guidance for practice as well as future research on the recommendation of exercises for older adults with AD to improve postural stability and as a result reduce falls.

## Figures and Tables

**Figure 1 ijerph-19-10350-f001:**
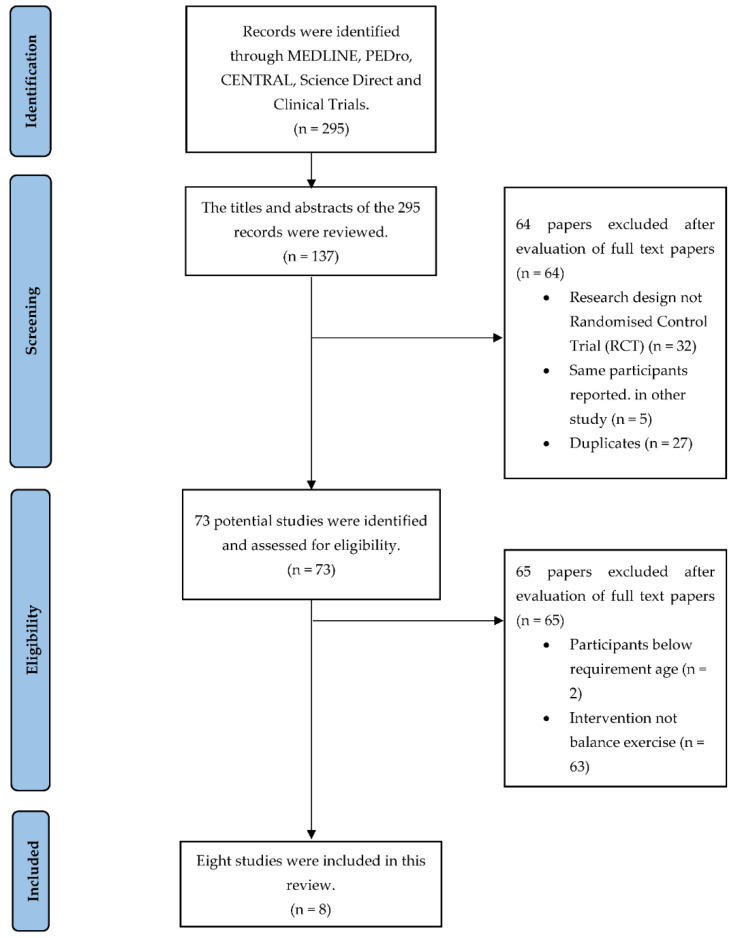
PRISMA flow diagram.

**Table 1 ijerph-19-10350-t001:** PEDro scores of included studies.

Study	Random Allocation	Concealed Allocation	Groups Similar at Baseline	Participant Blinding	Therapist BLINDING	Assessor Blinding	<15% Dropouts	Intention-to-Treat Analysis	Between-Group Difference Reported	Point Estimate and Variability Reported	Total (0 to 10)
Binns et al. (2020) [7]	1	0	1	0	0	1	0	0	1	0	4
Pitkälä et al. (2013) [21]	1	1	1	0	0	0	1	1	1	1	7
Öhman et al. (2016) [18]	0	0	1	0	0	0	0	1	1	0	3
Toots et al. (2019) [12]	1	1	1	0	1	1	0	1	1	0	7
Lamb et al. (2018) [20]	1	1	1	0	0	0	0	1	1	1	6
Roach et al. (2011) [22]	1	0	1	0	0	1	0	0	1	0	4
Santana-Sosa et al. (2008) [23]	0	0	1	0	0	1	1	0	1	0	4
Rolland et al. (2007) [24]	1	0	1	0	1	1	1	1	1	0	7

Abbreviations: 0 = no/not addressed/not applicable, 1 = yes.

**Table 2 ijerph-19-10350-t002:** Summary of included studies.

Study	Design	Participants	Intervention (Type of Exercise)	Time, Frequency, Intensity	Outcome Measures
Binns et al. (2020) [7]	RCT	n = 23 Age (yr) = 71 (mean) Gender = 6 M, 17 F ADAS-Cog11 = 16.2 (SD 8.2) Set = long term care	Exp = CogEx (aerobic, balance and strengthening exercise) Con = CST	1 h, 2x/week, 7 weeks Exp intensity: NR	MoCA 15-GDS QOL-AD ADAS-Cog11 Brief BESTest SPPB Follow up = 1 to 7 weeks
Pitkälä et al. (2013) [21]	RCT	n = 210 Age (yr) = 77.7 (SD 5.4) Gender = 129 M, 81 F MMSE = 17.8 (SD 6.6) Set = community	Exp = Group 1: Home based exercise Group 2: Group based exercise (endurance, balance, strength training and brain training)	1 h, 2x/week for 12 months Exp intensity: NR	MMSE FIM SPPB Follow up = 24 months
			Con = Community care (oral and written advice on nutrition and exercise)		
Öhman et al. (2016) [18]	RCT	n = 194 Age (yr) = 78 (SD 5.25) Gender = 119 M, 75 F MMSE = 22.9 (SD 3.6) Set = community	Exp = Group 1: Home based exercise. Group 2: Group based exercise (endurance, balance, and strength training)	1 h, 2x/week, 12 months Exp intensity: NR	MMSE CDR FIM Follow up = 3, 6, 12 months
			Con = no intervention		
Toots et al. (2019) [12]	RCT	n = 186 Age (yr) = 85.1 (SD 7.1) Gender = 141 M, 45 F MMSE = 14.9 (SD 3.5) Set = long term care	Exp = HIFE (lower limb strength, balance and mobility) Con = Interesting topic (local wildlife, seasons and holidays)	8 to 12 repetitions 45 min, 2x/week, 12 months Exp intensity: moderate to high	MMSE Barthel Activity Daily Living (ADL) Index 15-GDS BBS Follow up = 6, 12 months
Lamb et al. (2018) [20]	RCT	n = 494 Age (yr) = 78.4 (SD 7.6) Gender = 301 M, 193 F ADAS-Cog11 = 21.6 (SD 8.7) Set = community	Exp = aerobic and strength exercise Con = counseling, symptomatic treatment and advice on physical activity	60−90 min, 2x/week in group), additional 1 h/week (at home), 12 months Exp intensity: progressive aerobic (25 min), strength (20 reps, 3 sets); both moderate to high intensity	sMMSE QOL-AD ADAS-Cog 11 6MWT Follow up = 6, 12 months
Roach et al. (2011) [22]	RCT	n = 130 Age (yr) = 88.2 (SD 6.13) Gender = not stated MMSE = 10.2 (SD 7.6) Set = long term care	Exp = Group 1: aerobic, strength, flexibility, balance Group 2: aerobic group	15−30 min, 5x/week, 4 months Exp intensity: aerobic (between 10 to 30 min); strength & balace (low to high intensity, 2−3 reps progressed to 7−9 reps)	MMSE ACIF 6MWT
			Con = one-on-one conversation		
Santana-Sosa et al. (2008) [23]	RCT	n = 16 Age (yr) = 76.0 (SD 4.0) Gender = 6 male, 10 female MMSE = 20.45 (SD 4.0) Set = long term care	Exp = aerobic, flexibility, joint mobility exercise, resistance and balance/coordination training. Con = routine nursing/medical care.	75 min, 3x/week, 3 months Exp intensity: aerobic (light-without breathlessness); stretching (gentle); resistance (3 sets, 15 reps, medium resistance band)	MMSE Katz ADL score Barthel Activity Daily Living (ADL) Index Tinetti Scale 8-foot up & go test
Rolland et al. (2007) [24]	RCT	n = 134 Age (yr) = 83.0 (SD 7.4) Gender = 33 male, 101 female MMSE = 8.8 (SD 6.6) Set = long term care	Exp = aerobic, strength, balance and flexibility exercise. Con = routine medical care	60 min, 2x/week, 12 months Exp intensity: aerobic (light progressed to moderate breathlessness)	MMSE Katz ADL score 6-m walking speed TUG One-leg balance test MNA NPI MADRS

Abbreviations: ACIF = Acute Care Index of Function, ADAS-Cog11 = Alzheimer’s Disease Assessment Scale—Cognitive, BBS = Berg Balance Scale, Brief BESTest = Brief Balance Evaluation Systems Test, CDR = Clinical Dementia Rating, CogEx = Cognitive Stimulation Therapy with strength and balance exercise, Con = control group, CST = Cognitive Stimulation Therapy, Exp = experimental group, FIM = The Functional Independence Measure, HIFE = High-Intensity Functional Exercise Programme, MADRS = Montgomery Asberg Depression Rating Scale, MNA = Mini-Nutritional Assessment, MMSE = Mini Mental State Examination, MoCA = Montreal Cognitive Assessment, NPI = Neuropsychiatric Inventory QoL-AD = Quality of Life Alzheimer’s Disease, sMMSE = standardise Mini Mental State Examination, SPPB = Short Physical Performance Battery, TUG = Timed Up and Go 6MWT = 6 min walk test, 15-GDS = 15 items Geriatric Depression Scale, set = setting, reps = repetitions, NR = Not reported, x/ = times per.

**Table 3 ijerph-19-10350-t003:** Results to measure falls or postural stability of the included studies.

Study	Design	Target Group	Outcome Measures	Summary Findings
Binns et al. (2020) [7]	RCT	Age (yr) = 71 (mean) (n = 23)	Brief BESTest *	Brief BESTest change, 1.3 [95% CI −1.5 to 4.2] in the CST group and 1.0 [95% −1.1 to 3.1] in the CogEx
			SPBB*	SPBB change, 0.6 [95% CI −0.9, 2.0] in the CST, and −0.2 [95% CI −1.0 to 0.6] in CogEx
Pitkälä et al. (2013) [21]	RCT	Age (yr) = 77.7 (SD 5.4) (n = 210)	FIM *	6 months FIM change, −6.5 [95% CI, −4.4 to −8.6] in the HE, −8.9 [−6.7 to −11.2] in the GE, and −11.8 [95% CI, −9.7 to −14.0] in the CG, mixed effect model; P= 0.003 The difference between HE and CG at 6, *p* = 0.001 The difference between GE and CG at 6, *p* = 0.07 12 months FIM change, −7.1 [95% CI, −3.7 to −10.5] in the HE, −10.3 [95% CI, −6.7 to −13.9] in the GE, and −14.4 [95% CI, −10.9 to−18.0] in the CG, *p* = 0.15 The difference between HE and CG at 12 (mixed effect model), *p* = 0.004 The difference between GE and CG at 12, *p* = 0.12 No significant within group effect
			SPPB	The SPPB scores do not significantly affect mobility
Öhman et al. (2016) [18]	RCT	Age (yr) = 78 (SD 5.25) (n = 194)	FIM *	6 months FIM change, –3.3 [95% CI, –1.5 to –5.2] in the EG, and –8.9 [95% CI, –5.2 to –12.7] in the CG, *p* = 0.003 12 months FIM change, –2.7 [95% CI, –0.5 to –4.9] in the EG and –10.1 [95% CI, –7.0 to –13.3] in the CG, *p* < 0.001
			Fall rate *	Fall rate mild AD, IRR 0.65 [95% CI, 0.42 to 1.01], *p* = 0.055 Fall rate advanced AD, IRR 0.47 [95% CI, 0.37–0.60], *p* < 0.001
Toots et al. (2019) [12]	RCT	Age (yr) = 85.1 (SD 7.1) (n = 186)	BBS Fall rate	6 months BBS, IRR 1.3 [95% CI, 0.8 to 2.2], *p* = 0.303 Fall rate, IRR 0.9 [95% CI, 0.5 to 1.7], *p* = 0.838 12 months BBS, IRR 1.3 [95% CI, 0.7 to 2.4], *p* = 0.286 Fall rate, IRR 0.9 [95% CI, 0.5 to 1.6], *p* = 0.782
Lamb et al. (2018) [20]	RCT	Age (yr) = 78.4 (SD 7.6) (n = 494)	6MWT *	6 weeks: The distance in 6MWT change, 18.1 m [95% CI, 11.6 m to 24.6 m, *p* = 0.001 (change in distance was not reported in 6 and 12 months follow up)
			Fall rate	12 month IRR 1.1 [95% CI, 0.8 to 1.6], *p* = 0.69
Roach et al. (2011) [22]	RCT	Age (yr) = 88.2 (SD 6.13) (n = 130)	6MWT *	6MWT mean scores of participants in: EG increased 29.5%, from 163.90 m (SD 78.97) to 212.20 m (SD 137.54) Walking group increased 23.2% from 183.28 m (SD 83.60) to 225.83 m (SD 169.47) Conversation group increased 7.1% from 151.31 m (SD 64.22) to 162.00 m (SD 113.85) Improved but not statistically significant
Santana-Sosa et al. (2008) [23]	RCT	Age (yr) = 76.0 (SD 4.0) (n = 16)	Tinetti Scale *	Significantly improved in IG (*p* < 0.05); baseline <19, post intervention score mean 22 (SD 3) No improvement in CG (*p* > 0.05)
			8-foot up & go test *	Significant improvement in IG (*p* < 0.05) No changes in CG (*p* > 0.05)
Rolland et al. (2007) [24]	RCT	Age (yr) = 83.0 (SD 7.4) (n = 134)	6-m walking test *	Between group effect 6-m walking speed, significant improvement between 6 months (*p* = 0.01) and 12 months (*p* = 0.002)
			TUG	TUG, no significant improvement between 6 months (*p* = 0.68) and 12 months (*p* = 0.31)
			One-leg balance test	One-leg balance test, no significant improvement between 6 months (*p* = 0.47) and 12 months (*p* = 0.34)
				Whithin group effect was not reported for all measures

Abbreviations: BBS = Berg Balance Scale, Brief BESTest = Brief Balance Evaluation Systems Test, EG = Experimental Group, CG = Control Group, CI = Confidence Interval, CogEx = Cognitive Stimulation Therapy with strength and balance exercise, FIM = The Functional Independence Measure, GE = Group exercise HE = Home exercise group, HIFE = High-Intensity Functional Exercise Programme, IRR = Incidence Rate Ratio, SPPB = Short Physical Performance Battery, TUG = Timed Up and Go, 6MWT = 6 min walk test. * Post-intervention improvement.

**Table 4 ijerph-19-10350-t004:** Summary of intervention used in the experimental and control group.

Study	The Experimental Group (Intervention)	Control Group (Intervention)
Binns et al. (2020) [7]	CogEx: Aerobic, progressive strength and balance exercises	CST; planning, executing, naming and organizing tasks of (physical games, sounds, childhood. Food, current affairs, faces and scenes, associated words, being creative, categorizing objects, orientation, using money, number games, word games, team quiz)
Pitkälä et al. (2013) [21]	Physical exercises: Aerobic, strengthening, balance and exercise for improving executive functioning	Oral and written advice on exercise and nutrition
Öhman et al. (2016) [18]	Physical exercises: Aerobic, strengthening, balance and dual-tasking exercise	Advice on exercise, nutrition and regular treatment in the healthcare system
Toots et al. (2019) [12]	Physical exercises: Lower limb strengthening, balance and mobility	Structured interesting activities, including holidays, wildlife and seasons
Lamb et al. (2018) [20]	Physical exercises: Aerobic and strengthening	Counselling, symptomatic treatment and advice on physical activity
Roach et al. (2011) [22]	Physical exercises: aerobic, strength, balance and flexibility exercises.	One-on-one conversation on participants’ topic of interest
Santana-Sosa et al. (2008) [23]	Physical exercises: aerobic, strength, balance and flexibility exercises.	Routine nursing/medical care
Rolland et al. (2007) [24]	Physical exercises: aerobic, strength, balance and flexibility exercises.	Routine medical care

Abbreviations: CogEx = Cognitive Stimulation Therapy with strength and balance exercise, CST = Cognitive Stimulation Therapy.

## Data Availability

Not applicable.

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
