# Peer review of "Exercise to Improve Postural Stability in Older Adults with Alzheimer’s Disease: A Systematic Review of Randomized Control Trials"

_ijerph, 2022, doi:10.3390/ijerph191610350_

Round 1

Reviewer 1 Report

This paper is a review paper on the improvement of postural disability and reduce falls in older adults with Alzheimer's Disease. There are problems needed to be clarified and addressed.

The main drawback is that the authors set up inclusion criteria and eventually only five papers were retained and analyzed. This sample was too small for a review paper.   The authors need to explain how such a small sample is meaningful to address the issues in this study.  The small sample size is linked to issues related to the eligibility or inclusion criteria of the literature. An example is the age of the participants. Line 79-80, participants aged 60 yrs (Asian) and 65 yrs (European) who had been  ..."  Aged 60 and 65 yrs exactly? or 60 and 65 yrs or older? Why the age or age limit for Asian and European are different? How were these age determined? Why not 70, 75 yrs, etc?  

Line 83, "Positive trials or studies with no improvements in outcomes were included in the review."  What is positive trials? Were there negative trials?

Does the sentence implies that studies with improvements in outcomes were excluded? Why? 

Line 99, "<less than 15% dropouts"  Why 15%?

Line 104, "Two reviewers"  Were they external reviewers? or two of the co-authors?    The same question for the "One reviewer" in Line 109.

Line 110, "A meta-analysis was not conducted." You don't need to mention what you didn't do. 

Line 116, "...was less than 65 yrs.." Was this for European or Asian? 

The quality of Figure 1 is poor. The lines with arrow head are either too long or too short which make the readability of this figure poor.

Table 1, What is "Total" in the last column?  If it is the summation of the number of "Y" in the same row, I suggest you use 1 and 0 to replace Y and N. This will make the "Total" more sense. 

Poor English:

Line 86, "The comparison of not balance...." This sentence needs to be rewrite.

Author Response

Many thanks for all the comments to improve the manuscript. Please see the attachment for the feedback. The feedback were highlighted.

Reviewer 2 Report

This article reviews the effects of exercise intervention on postural stability and fall reduction in AD. The content is very interesting as a theme, but at present, there are too few studies to be used as a references, so it is meaningful as a review to summarize.

Author Response

Many thanks for all the comments to improve this manuscript. Please see attachment for the feedback. The feedback was highlighted. 

Reviewer 3 Report

In this manuscript, the authors conducted a systematic review for older adults with AD. Overall, the article seems very interesting to me and is timely. However, the authors should provide more information about the methodology, discuss their results more concisely and briefly,

You need to arrange the order of the references. It currently starts at number 17 (line 30). And, please refer to the "MDPI Style Guide" for references.

As mentioned in "limitation", it is difficult to suggest conclusions from a review of five randomized studies. Therefore, it is considered that supplementation (additional thesis data and systematic arrangement of existing thesis) is necessary.

I present my comments and suggestions for changes in relation to the following parts of the article.

(Line 47) It seems that "A consequence of falls, hip fractures, may impact ...." should be modified like "A consequence of falls may impact ...".

(Line 50-52, Line 79-83, Line 101-102, Line 131-132) These sentences seems to have grammatical problems. I think you need to correct the part where there is a problem with the grammar.

(Line 72) "... publisehd from 2010 to 2021." The period in which the electronic literature search was performed is different from the abstract (between November 2020 and January 2021). Please check.

(Line 108-109) I don't understand exactly what a customised data extraction table is. A specific explanation is required.

(Line 122-129) This sections are not reader-friendly.

(Line 162-163) "RCT or QCT" Please include the full terms before using an abbreviation.

(Line 186) In terms of readability, I think Figure 1 needs to be modified. In the process of identification, screening, eligibility, and included, the calculation of the number of articles included or excluded is not accurate.

(Page 5) In terms of readability, some modifications to Table 1 are needed.

(Page 6-7) In terms of readability, the "Frequency/Intersity and Type" and "Outcome measures" part of Table 2 needs to be corrected. In the captions below, "Notes" should be modified to "Abbreviations".

(Page 8-9) In terms of readability, some modifications to Table 3 are needed.

Author Response

Many thanks for the comments to improve this manuscript. Please see attachment for the feedbacks. The feedbacks were highlighted.

Round 2

Reviewer 1 Report

I have no further comments.

Author Response

Many thanks for your time and expertise reviewing this manuscript.

Reviewer 2 Report

The fact that the number of analytical treatises has reached eight is considered to be meaningful to some extent.

Author Response

Many thanks for your time and expertise in reviewing the manuscript.

Reviewer 3 Report

In my opinion, a separate document format with each individual reviewer is more appropriate than a single unified document format containing all reviewers' comments and responses.

Also, please provide the final manuscript file with the revised contents reflected so that the reviewers can check the contents more easily.

Round 3

Reviewer 3 Report

Overall, this manuscript is very well written and organized.

Congratulations for the work done.

Author Response

Dear reviewer,

Many thanks for the comments and feedback for the improvement of this paper. We highly appreciated your support.

Kind regards,

Normala Mesbah